

# Solar and lunar tides in noctilucent clouds as determined by ground-based lidar

Jens Fiedler[1] and Gerd Baumgarten[1]

[1]Leibniz-Institute of Atmospheric Physics, Rostock University, Kühlungsborn, Germany

*Correspondence to:* Jens Fiedler (fiedler@iap-kborn.de)

**Abstract.**

Noctilucent clouds (NLC) occur during summer from mid to high latitudes. They consist of nanometer sized ice particles in an altitude range from 80 to 90 km and are sensitive to ambient temperature and water vapor content, which makes them a suitable tracer for variability on all time scales. The data set acquired by the ALOMAR RMR-lidar covers 21 years and is investigated regarding tidal signatures in NLC. For the first time solar and lunar tidal parameters in NLC were determined simultaneously from the same data. Several NLC parameters are subject to persistent mean variations throughout solar as well as lunar day. Variations with lunar time are generally smaller compared to variations with solar time. NLC occurrence frequency shows the most robust imprint of the lunar semidiurnal tide. Its amplitude is about 50 % of the solar semidiurnal tide, which is surprisingly large. Phase progressions of NLC occurrence frequency indicate upward propagating solar tides. Below 84 km altitude the corresponding vertical wavelengths are between 20 and 30 km. For the lunar semidiurnal tide phase progressions vary symmetrically with respect to the maximum of the NLC layer.

## 1 Introduction

Noctilucent clouds (NLC) are a phenomenon of the mesopause region from mid to high latitudes. During summer temperatures fall below 150 K and cause the few ppmv of water vapor at these altitudes to freeze. The result are tiny ice particles which are observable by naked eye. First documented observations have been made during summer 1885 (Backhouse, 1885; Jesse, 1885; Leslie, 1885) and the altitude of the clouds was determined to about 83 km by optical triangulation (Jesse, 1896).

It has been shown that NLC consist of few tens of nanometer sized, aspherical ice particles with a number density of about hundred per cm$^3$ (e.g., von Cossart et al., 1999; Hervig et al., 2001; Baumgarten et al., 2002). NLC displays often show pronounced variability, which provides information about dynamic processes like horizontal wind and wave motions in their environment (Witt, 1962). Mesospheric ice particles are extremely sensitive to changes of atmospheric background parameters, like temperature and waver vapor, and have been demonstrated to respond to various scales of variability, ranging from seconds to years (e.g., Thomas et al., 1991; Kirkwood and Stebel, 2003; Merkel et al., 2003; DeLand et al., 2003; Kaifler et al., 2013; Russell III et al., 2014; Fiedler et al., 2017).

One particular scale of interest are variations with solar time (e.g., von Zahn et al., 1998; Chu et al., 2001; Fiedler et al., 2005; Gerding et al., 2013). Such oscillations have been found to be persistent even when epoch averaging over many years and



where attributed to impacts of atmospheric thermal tides (e.g., Fiedler et al., 2011; Stevens et al., 2017). Solar tidal oscillations are globally forced due to absorption of solar irradiance throughout the day. Prominent influence have diurnal and semidiurnal components, which are stimulated by absorption of solar radiation in the near-infrared bands of tropospheric water vapor and solar ultraviolet radiation by stratospheric ozone and mesospheric molecular oxygen, respectively (e.g., Chapman and Lindzen,

1970; Forbes, 1984).

The moon forces gravitational tidal signatures not only in Earth's oceans but also in atmosphere. Sabine (1847) published the first reliable identification of lunar tidal signatures in surface pressure data. The semidiurnal lunar tide is the most significant component and was found in several parameters even in the mesosphere and lower thermosphere like winds, temperatures and airglow emissions (e.g., Stening et al., 1987; Paulino et al., 2013; von Savigny et al., 2015). Studies of lunar tidal signatures

in NLC are sparse and mostly based on ground-based visual observations (e.g., Kropotkina and Shefov, 1975; Gadsden, 1985; Dalin et al., 2006). Only recently the lunar semidiurnal tide was identified in multi-year data sets of satellite instruments (von Savigny et al., 2017; Hoffmann et al., 2018). Ground- and satellite-based results partly differ from each other, especially for the lunar tidal amplitude.

In general, available observation methods of tides in NLC have different pros and cons. Ground-based visual data can only

be obtained around solar midnight hours where the lower troposphere is dark but sunlight still illuminates the upper mesosphere and is scattered at the ice particles. This hampers the identification of both solar and lunar tidal signatures. Ground-based lidars can cover the entire solar diurnal cycle and are thus able to identify solar and lunar tides. As they operate at fixed locations, the superposition of all tidal components overhead the instrument is measured, with the migrating components giving the distinct variations with solar time. Solar and lunar semidiurnal periods are close to each other (12.0 vs. 12.4 solar hours) and a

separation requires adequate sampling and data accuracy. Satellite instruments usually operate in sun-synchronous orbits and cover only few solar times during longer time periods, which hampers the identification of solar tidal signatures. They sample continuously and have less deterioration of the lunar signal by solar impacts, however, it takes one month to cover all lunar times.

In this study we will extract for the first time solar and lunar tidal signatures in NLC simultaneously from a multi-year data

set obtained by ground-based lidar. Moreover, phase progressions will be addressed by investigation of altitude resolved tidal parameters.

## 2   Data analysis

We use data obtained by the Rayleigh/Mie/Raman-(RMR-) lidar at the ALOMAR research station in Northern Norway (69° N, 16° E). The lidar is in regular operation since 1997 and is intensely used during the summer months for NLC detections

whenever weather conditions permit. Because of the technical setup NLC are detectable during all local times, even during the highest solar elevation angles around 44°. NLC above ALOMAR usually occur between begin of June and mid August, leading to our definition of the season length from day of year (DoY) 152 (1 June) to 227 (15 August). The data set covers





6400 measurement hours during 21 seasons, a subset of 3100 hours contains NLC. Hence, the mean probability to observe NLC at this location is ∼48 %.

The lidar transmitter emits light at 532 nm wavelength with approx. 50 MW power per laser pulse. Part of it is backscattered by air molecules and NLC particles and detected by the receiver, together with sunlight scattered by the atmosphere.

After background subtraction the volume backscatter coefficient as function of altitude $\beta_{\mathrm{NLC}}(z)$ is calculated. Values larger than zero indicate the presence of aerosol particles and are a measure of the cloud brightness. The following NLC parameters are extracted: occurrence frequency (OF), centroid altitude, maximum backscatter coefficient of the altitude profile ($\beta_{\max}$), and total backscatter coefficient of the column ($\beta_{\mathrm{tot}}$). For details the reader is referred to Fiedler et al. (2009). To maintain homogeneous conditions during 21 seasons, the data were pre-integrated in time and altitude for about 15 min and 150 m.

In a next step, the measurements were sorted by local time, i.e. the individual pre-integrations were accumulated and averaged in their corresponding hourly time slots. This method is usually called superposed epoch analysis and was applied for solar as well as lunar times separately. Lunar, like solar, time follows from the azimuth position of the celestial body relative to the observer. For a given solar time Moon ephemerides were calculated using the 'PyEphem' library (https://pypi.python.org/pypi/pyephem). To extract tidal information from the temporal variations of NLC parameters, least-

square fits of the sum of sinusoidal functions with periods of 24, 12, 8, 6 hours to the hourly mean values (solar as well as lunar) were performed. Confidence intervals of the estimated fit parameters were calculated with the bootstrap method by resampling the data set within the uncertainties of the means (e.g., Efron and Tibshirani, 1993). The mean NLC parameters are randomly diversified within their error bars (1000 times for each hour), which results in an equivalent number of time series for which the fits are determined. Finally, the statistics of the distribution of each fit parameter is calculated, resulting in a mean value

and its error.

## 3   Results and discussion

### 3.1   Variations with solar time

Fig. 1 shows the mean variations of NLC occurrence frequency and brightness with altitude and time. The plots contain 6400 **Fig.1** measurement hours and were composed using each single altitude profile of the NLC brightness from 1997 to 2017. NLC

above ALOMAR can virtually exist in the entire altitude range between 78 and 90 km. They occur most often and have the largest vertical extent between midnight and 6 local solar time (LST), which was attributed to thermal tides at 83 km altitude (Fiedler et al., 2011). The altitude of maximum occurrence decreases by about 1 km during the morning hours. A second and weaker occurrence maximum is visible around 15 LST. The clouds reach their maximum brightness between 3 and 9 LST, which is 3 hours later compared to the occurrence maximum. Nevertheless relatively strong (brighter) clouds contribute to

this occurrence maximum. The secondary occurrence maximum, however, is caused only by fainter (dimmer) clouds. The altitude structure of brightness mirrors the growth-sedimentation scenario of NLC particles. They nucleate at low temperatures in the mesopause region around 88 km, grow in size by the uptake of water vapor and decrease in altitude due to selective turbulent diffusion and gravitation. The observed brightness depends strongly on particle size ($\propto \mathrm{r}^6$). Temperature increases





with decreasing altitude, which causes the ice particles to sublimate. This leads to a sharp brightness decrease at the lower border of the particle existence range.

We like to point out that from measurements at one location one cannot prove that observed local time dependent features are caused by tides. However, the persistence of features in our 21 year data set shown in Figure 1 is a strong hint for atmospheric
tides. Imprints of variability sources uncorrelated to solar time, like gravity waves, should cancel out on these multi-year time scales. Furthermore we observe a superposition of all existing tidal modes at a given time and cannot differentiate between migrating and nonmigrating parts.

## 3.2 Simultaneous solar and lunar tidal variations

We investigate the mean local time dependence of NLC parameters using a representative brightness value for each altitude
profile. This method was applied earlier for our data set and is commonly used for ground-based lidars as well as satellite instruments. We determine for each altitude profile the value of the maximum brightness $\beta_{\max}$ as well as the integrated brightness $\beta_{\mathrm{tot}}$ over all altitudes. Local time variability of NLC parameters depends on cloud brightness and its observation is additionally impacted by instrument sensitivity (e.g., Fiedler et al., 2005, 2011). For these reasons we usually apply a minimum brightness limit. In this study, however, we aim on the identification of weak lunar signals embedded in a larger background of solar
variability. The frequency distribution of brightness values satisfies an exponential law (cf. Fig. 1 in Fiedler et al. (2017)) with a high occurrence of dim clouds. As a result, the application of our long-term brightness limit ($\beta_{\max} > 4 \times 10^{-10}\,\mathrm{m}^{-1}\,\mathrm{sr}^{-1}$) reduces the NLC detections by 48 % from 3100 to 1600 hours. Sensitivity tests showed that data reductions of such extent prevent a reliable extraction of the lunar signal from our data set. It turned out that a limitation to the core of the season is better suited as it restricts to stable summer conditions with high NLC occurrences. Therefore we choose a time period where the
daily NLC occurrence frequency exceeds the seasonal mean value, which is roughly the case between DoY 170 (19 June) and 210 (29 July); cf. Fig. 3 in Fiedler et al. (2009) and Fig. 2 in Fiedler et al. (2017). This reduces the NLC detections by 35 %.

The resulting solar and lunar time variations are shown in Fig. 2 . The plots contain 3450 measurement and 2030 NLC **Fig.2**
hours from 1997 to 2017. Throughout the solar day the results match the variations seen in Figure 1. We again find the highest NLC occurrence between midnight and 6 LST and a weaker maximum around 15 LST. The maximum brightness is observed
between 3 and 8 LST. Highest altitudes are reached around midnight and 14 LST, the mean altitude variation during solar day is ∼1.1 km. The daily variations of brightness and altitude are anti-correlated (r = −0.84), i.e. brightest clouds occur at lowest altitudes and vice versa, which fits the above-mentioned growth-sedimentation scenario of mesospheric ice particles.

Such distinct variations throughout the solar day were first observed 1997 at ALOMAR and have been found by ground-based lidar at other locations as well (e.g., von Zahn et al., 1998; Chu et al., 2001; Fiedler et al., 2005; Gerding et al., 2013).
Satellites mostly operate in sun-synchronous orbits and are thus not able to cover NLC local time variations, with some exceptions (e.g., Stevens et al., 2009; DeLand et al., 2011). NLC variability during solar day was also investigated by models (e.g., Stevens et al., 2010, 2017; Schmidt et al., 2017). In general, maximum values for occurrence and brightness are found in the first half of the solar day, which is attributed to temperature tides and tidal variations in background water vapor.



Figure 2 shows also the NLC parameters as function of local lunar time (LLT). We find variations in all parameters, most intense for occurrence frequency. The signatures in altitude and brightness are less pronounced and better visible during the first half of the lunar day. NLC occur most often around 3 and 11 LLT. The highest altitude is reached around 2 LLT, which is connected with a minimum in brightness. Variations with LLT are smaller compared to that with LST. The relative variation

$\Delta = (\max - \min)/\text{mean}$ during solar (lunar) day is 60.9 % (21.5 %) for occurrence frequency, 1.4 % (0.4 %) for altitude, 74.4 % (15.8 %) for maximum brightness, and 92.8 % (25.7 %) for total brightness.

To compare the impact of solar and lunar tides on NLC we extracted amplitudes (A) and phases (P) of fits up to the $4^{th}$ harmonic of the day to the data. The results are listed in Table 1 . We note the only moderate correlation coefficients for altitude **Tab.1** and especially brightness variations with lunar day, indicating these parameters to be additionally impacted by other sources of variability. Solar tidal variations are dominated by diurnal and semidiurnal periods. Amplitude ratios $A_{24}/A_{12}$ are 1.8, 2.1 and 2.5 for occurrence frequency and brightness (maximum, total), respectively. For altitude both periods identically contribute to solar time variations.

Investigations of NLC regarding lunar tidal signatures are sparse and mostly based on ground-based visual observations. Such observations are limited to a couple of hours around midnight because of the illumination conditions of mesospheric altitudes by the sun. The results partly differ from each other and show lunar variations of the NLC occurrence with amplitudes from 4 to 30 % in a months period (e.g., Kropotkina and Shefov, 1975; Gadsden, 1985; Dalin et al., 2006). Only recently lunar tides in NLC were identified in multi-decade data sets of SBUV (Solar Backscattered Ultraviolet) satellite instruments (von Savigny et al., 2017). The authors found clear lunar semidiurnal tidal signatures in NLC occurrence frequency, albedo, and ice water content. For the northern hemisphere (55°–75° N) they extracted a relative amplitude $A_{12}(\text{rel})$ of 5.2 % and a phase $P_{12}$ of 3.3 h LLT for the NLC occurrence frequency, which is so far the only value determined by instrumental observations. The present study is the first identification of lunar tidal signatures in ground-based lidar observations. Our values of $A_{12}(\text{rel}) = 6.8$ % and $P_{12} = 2.0$ h LLT are in good agreement with von Savigny et al. (2017). From visual NLC observations published by Gadsden (1985) follow maximum NLC occurrences around 3 h LLT (cf. discussion in von Savigny et al. (2017)). Thus three independent data sets show occurrence maxima between 2 and 3 h LLT, which hints to a robust determination of the corresponding semidiurnal lunar tide.

von Savigny et al. (2017) suggested temperature variations in the mesopause region as main driver of lunar tidal signatures in NLC. They investigated 7 years of satellite temperature data from the MLS (Microwave Limp Sounder) instrument and found at 83 km altitude consistent features with respect to lunar NLC variations, namely minimum temperatures from 1 to 4 h LLT. Hoffmann et al. (2018) analyzed 9 years of data from the SOFIE (Solar Occultation for Ice Experiment) satellite instrument and found temperature variations with LLT as well. In general temperature variations with lunar time are very small with maximum amplitudes of about 0.2 K.

Brightness related parameters (albedo, ice water content) were determined by von Savigny et al. (2017) to be $A_{12}(\text{rel}) \sim 6$ % and $P_{12} \sim 3$ h LLT. Hoffmann et al. (2018) found in SOFIE data $A_{12}(\text{rel}) = 2.5$ % and $P_{12} = 2.5$ h LLT for the ice water content in the northern hemisphere. Our values of $A_{12}(\text{rel}) = 2.1$ % and $P_{12} = 4.4$ h LLT for brightness ($\beta_{\text{tot}}$) match reasonably the satellite data. We notice, however, that the semidiurnal component is the weakest of all extracted harmonic components for our NLC



brightness (cf. Table 1). The superposition of all four oscillations results into a brightness minimum around 2 h LLT, compared to a maximum in the satellite observations. Hoffmann et al. (2018) also extracted lunar tidal NLC altitude variations, which were determined to $A_{12}(\mathrm{abs}) = 60\,\mathrm{m}$ and $P_{12} = 2.2\,\mathrm{h}$ LLT. This fits remarkably our values of $A_{12}(\mathrm{abs}) = 80\,\mathrm{m}$ and $P_{12} = 0.9\,\mathrm{h}$ LLT. A difference between satellite results and the present study is the phasing of altitude to brightness. Whereas Hoffmann

et al. (2018) find them to be in phase (clouds with larger ice water content are at higher altitudes), our results show the common anti-phase behavior known from solar tidal variations (cf. Figure 2). When only the semidiurnal components are taken into account, the lunar phases of altitude and brightness ($\beta_{\mathrm{tot}}$) differ by about 3.5 hours, which is midway between in phase and anti-phase. We make aware that the statistical significance for our brightness fits is relatively low and thus this particular result should not be overestimated, see also below.

From Table 1 we find solar tidal amplitudes to be always larger than lunar ones, which is expected. For example, the ratios of semidiurnal solar to lunar amplitudes are $\sim$1.9 for occurrence frequency, $\sim$4.1 for altitude, and $\sim$7.5 for brightness. Dalin et al. (2017) extracted this ratio to $\sim$7.7 for horizontal winds from radar measurements in the mesopause region at 68° N. For our data set we notice that all four extracted harmonics contribute to the observed lunar tidal behavior, different for the NLC parameters. For occurrence frequency and altitude the semidiurnal component is dominating, whereas it is the weakest one for

brightness. From models a diurnal lunar tide is anticipated to be significantly smaller compared to the semidiurnal lunar tide (e.g., Chapman and Lindzen, 1970; Pedatella et al., 2012). Stening (1989) found a diurnal modulation of the lunar tide and suggested this to be caused by interactions with solar tides during upward propagation of the lunar tide, cf. also Stening and Vincent (1989). Concerning the 8-h and 6-h LLT oscillations we see no resilient reason to attribute it to be directly caused by the Moon and will estimate the robustness of extracted lunar oscillations in the following chapter.

## 3.3   Reliability of tidal parameters

For analysis of simultaneous solar and lunar tidal variations we use 41 days in the core of the NLC season (DoY 170–210). As our measurements cover many solar times and solar tidal amplitudes are large, the actual distribution of measurements times during each year might cause a residual impact on the extracted lunar amplitudes (sampling issue). This would introduce a systematic error on top of the statistical error of the lunar tidal parameters. We investigated this topic using the following

simulations.

At first the mean solar time dependencies for NLC occurrence frequency, altitude and brightness were reconstructed using amplitudes and phases of the harmonic solar periods. Then the constructed NLC parameter value at the solar time of each actual measurement was taken and assigned to the corresponding lunar time (method 1). This results into mean lunar time dependencies for the NLC parameters which should be ideally flat curves representing the mean values of the NLC parameters.

We find deviations from this ideal case indicating residual impacts from solar onto lunar tidal parameters (thick green curves in left panels of Fig. 3 ). For occurrence frequency enhanced values are visible from 4–6 LLT and from 11–16 LLT. For altitude   **Fig.3** enhanced values show especially from 10–12 LLT and decreased values from 16–18 LLT. The measured hourly mean values with respect to lunar time are also shown for reference (blue symbols and curves in left panels of Fig. 3). Comparing these curves with the simulations we find NLC occurrence frequency as well as altitude only be small impacted during lunar morning,




whereas during lunar afternoon solar impacts are large compared to the measured values. For NLC brightness the situation is worse. Here solar impacts reach larger values during extended lunar time periods, even during lunar morning.

We also calculated a second simulation (method 2). For this purpose, artificial solar times were randomly generated with the total number matching the number of measurements. Then, like in the first simulation, the constructed NLC parameter value

at each artificial generated solar time was taken and assigned to the corresponding lunar time. This procedure was executed several times. The results are the thin green curves in the left panels of Fig. 3. Again we find residual solar tidal impacts, however, they do not exceed the ones from the first simulation.

In general, the actual sampling of the measurements concerning solar time impacts the extracted lunar time dependence of NLC parameters. This impact is smallest for occurrence frequency, moderate for altitude, and largest for brightness. We notice

that 1.7 times more data are available for determination of occurrence frequencies (entire measurement time) compared to NLC layer parameters (only NLC measurement time). Thus, enhanced lunar amplitudes especially for the higher harmonics (8- and 6-h lunar periods) of altitude and brightness might be caused by an insufficient amount of data, although the data set covers 21 seasons.

For completeness we performed the same investigations regarding lunar residual impacts on solar tidal parameters. The

results are shown in the right panels of Fig. 3 and indicate only negligible effects.

### 3.4   Altitude dependence of tidal parameters

Now we study altitude resolved tidal parameters. For this purpose the altitude range between 80 and 88 km was divided into 8 slices of 1 km extent each (cf. Fig. 1). For each altitude slice amplitudes ($A_{24}$, $A_{12}$) and phases ($P_{24}$, $P_{12}$) of diurnal and semidiurnal harmonic oscillations were extracted. The result for solar tides is shown in Fig. 4 . Amplitudes of the occurrence   **Fig.4**

frequency reach values up to 10 %. The lower half of the altitude range is dominated by the diurnal component, being partly 1.9 times stronger compared to the semidiurnal component. At higher altitudes the amplitudes of both components decrease and are nearly identical. The brightness shows roughly a similar behavior. The diurnal component dominates by a factor up to 2.2. However, the altitude dependences of $A_{12}$ are different for occurrence frequency and brightness. While the former has its maximum at 84.5 km, the one of the brightness is monotonically decreasing with increasing altitude.

Phases for both tidal components of occurrence frequency and brightness decrease continuously with altitude, as is expected for upward propagating tides, with only one exception ($P_{12}$ of brightness). We notice the existence of two altitude ranges with different phase progressions, separated at about 84 km. From the slopes we determined corresponding vertical wavelengths $\lambda_z$ which are also shown in Figure 4. For the occurrence frequency, vertical wavelengths in the lower altitude range are $-31$ km ($P_{24}$) and $-21$ km ($P_{12}$). Values increase in the upper altitude range to $-125$ km ($P_{24}$) and $-56$ km ($P_{12}$).

For brightness we extract vertical wavelengths of $-68$ km ($P_{24}$) and $-17$ km ($P_{12}$) below 84 km altitude. Above this limit the progression of $P_{24}$ increases substantially, whereas progression of $P_{12}$ tends to change sign. This phase behavior, however, is accompanied by small amplitude values and might lack robustness. We notice that vertical wavelengths connected with diurnal phases have generally larger absolute values compared to that of semidiurnal phases.





Hough modes of classical tidal theory are distinguished by their vertical wavelength (Chapman and Lindzen, 1970). Wavelengths between 17 and 21 km as they were observed for the semidiurnal tide below 84 km correspond to higher order Hough modes H(2,9)–H(2,11). For the diurnal tide we find solely wavelengths $\geq 31$ km which indicates negative Hough modes. We note, however, that the excitation intensity of modes decreases towards higher latitudes according to linear theory. Thus at

69° N nonlinear wave interactions might play a major role.

To our knowledge, vertical phase progressions in NLC have never been published so far. Thus we compare our results to other parameters like temperature and horizontal winds measured in the summer mesopause region. Lübken et al. (2011) investigated thermal tides at Davis (69° S) by means of a resonance lidar. They found during January 2011 downward progressing $P_{24}$ with a vertical wavelength of $-30$ km between 84 and 89 km altitude. $P_{12}$ shows the opposite behaviour, namely upward progression

(their Figure 3). Murphy et al. (2006) published a climatology of tides in the Antarctic mesosphere determined by radar wind measurements. They found between December and February in the 80 to 86 km range vertical wavelengths from $-37$ km to $-55$ km for $P_{12}$ (migrating), but also values marked as *large*. Vertical wavelengths associated with $P_{24}$ are large or even positive, cf. their Figure 7. At mid-latitudes She et al. (2002) (Figures 2 and 3) found tidal temperature variations between 84 and 89 km altitude during summer corresponding to vertical wavelengths of $-27$ km ($P_{12}$) and $-19$ km ($P_{24}$) using a resonance lidar.

Again at mid-latitudes, from lidar temperature soundings by Kopp et al. (2015) follow vertical wavelengths of $-9$ km ($P_{12}$) and $-14$ km ($P_{24}$) around 85 km altitude in July (cf. their Figure 7). These numbers might not be representative, as their results show large variabilities during the summer period.

In general, solar tidal phases determined from our NLC observations show a consistent behavior indicating that corresponding vertical wavelengths are robust. The wavelengths fall within the range of values extracted from other measurements

published in the literature. We notice the variability of $\lambda_z$ values which might be caused by different time periods covered by the measurements (days to years) as well as the tidal variability itself. Changes in phase progressions around 84 km altitude could be caused by combined effects of tracer (ice particles) and background atmosphere. With increasing altitude the particle size decreases towards the mesopause at around 89 km where they nucleate. Simultaneously, turbulent mixing of the background atmosphere increases with altitude which impacts the microphysical properties of NLC particles. Baumgarten et al.

(2010) found a correlation between particle size and distribution width for mean sizes up to 40 nm, which are reached at an altitude of about 84 km. For larger particles (lower altitudes) the distribution width is roughly constant. Following this, one could speculate that tidal impacts might depend on the turbulent regime of the atmosphere.

We have shown that occurrence frequency is the most robust NLC parameter of our data set concerning lunar time variations. Therefore we investigated the altitude dependence of its semidiurnal component by applying the same procedure like for solar

time variations. The results are shown in Fig. 5 . The semidiurnal component maximizes shortly above the altitude of maximum **Fig.5** occurrence frequency. Phases vary between 1.5 and 4.4 h LLT, their progression is positive below 84 km and negative above. Corresponding vertical wavelengths are approx. $-27$ km and $+36$ km.

The altitude structure of the lunar semidiurnal tide in layered phenomena of the summer mesopause region was never studied so far. Paulino et al. (2013) found in temperature data of the SABER (Sounding of the Atmosphere Using Broadband Emission

Radiometry) satellite instrument both positive and negative phase progressions between 80 and 90 km altitude, depending on





latitude. They also identified several modes of the lunar semidiurnal tide, including nonmigrating, and suggested an interaction between lunar tide and other tides and/or waves at higher altitudes.

## 4 Conclusions

The 21 years data set of the ALOMAR RMR-lidar contains the largest NLC archive acquired by ground-based lidar and was investigated regarding solar and lunar tides in NLC. Distinct variations with solar as well as lunar time were found in several NLC parameters. This study represents the first identification of lunar tidal signatures in ground-based lidar observations. The results may be summarized as follows.

1. Throughout the solar day the highest NLC occurrence frequency was found between midnight and 6 LST, the brightness maximizes between 3 and 8 LST, and highest altitudes are reached around midnight and 14 LST. Throughout lunar day NLC occur most often around 3 and 11 LLT and the highest altitudes are reached around 2 LLT. Variations with lunar time are generally smaller compared to variations with solar time.

2. Solar time variations are dominated by diurnal and semidiurnal tidal components. For NLC occurrence frequency and brightness the diurnal component is roughly twice as large as the semidiurnal component. For NLC altitude both components identically contribute to the variability.

3. The relative amplitude of the lunar semidiurnal tide in NLC occurrence frequency is 6.8 % and the phase is 2.0 lunar hours. The lunar semidiurnal tide in NLC altitude has an amplitude of 60 m and a phase of 0.9 lunar hours. Both findings are in good agreement with results from satellite observations.

4. For the first time solar and lunar tidal parameters in NLC were determined simultaneously from the same data set. For occurrence frequency the lunar semidiurnal amplitude is approx. 50 % of the solar semidiurnal amplitude, which is surprisingly large.

5. We showed for the first time vertical resolved tidal parameters in NLC. For occurrence frequency phases of solar diurnal and semidiurnal components decrease with altitude, hinting for upward propagating tides. Corresponding vertical wavelengths are $-31$ km and $-21$ km below 84 km and larger above. Lunar semidiurnal phases stay in the range between 1.5 and 4.4 lunar hours and vary symmetrically with respect to the maximum of the NLC layer. Corresponding vertical wavelengths vary from approx. $-27$ km to $+36$ km.

6. Simulations of sampling effects showed that the distribution of lidar measurements in terms of solar time has impacts on the extracted lunar tidal parameters. In this way solar tidal variations cause residual variations in terms of lunar time. Such impacts are smallest for occurrence frequency, moderate for altitude, and largest for brightness. On the other hand lunar tidal variations have negligible impact on extracted solar tidal parameters. Following this, trend investigations of our NLC time series are not significantly affected by lunar tidal variations.





*Competing interests.* No competing interests are present.

*Acknowledgements.* We gratefully acknowledge the support of the ALOMAR staff in helping to accumulate the extensive data set of NLC observations. The observations were also supported by a large number of voluntary lidar operators. We thank Götz von Cossart for the excellent support in maintaining the lasers of the ALOMAR RMR-lidar. We are also grateful to Uwe Berger for hints improving the manuscript.



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





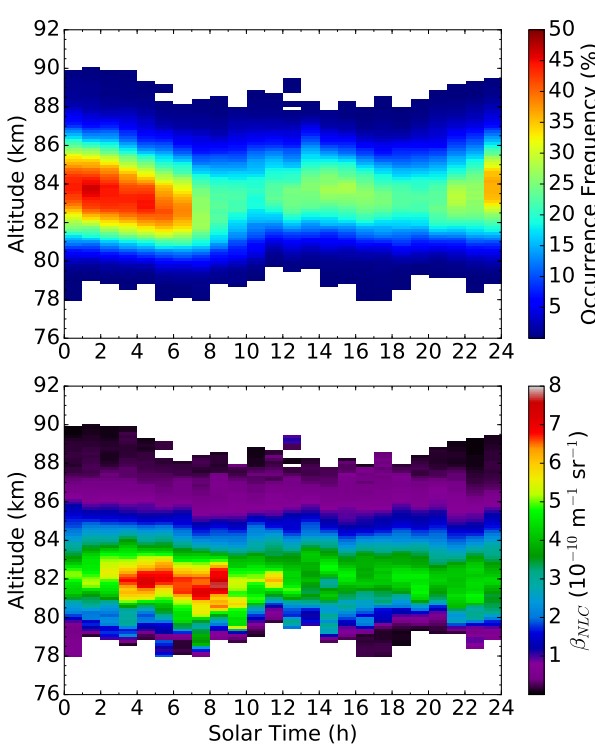

**Figure 1.** Mean altitude and solar time variations of NLC occurrence frequency (top) and brightness (bottom) from 1997 to 2017. The plots contain 6400 hours of lidar measurements between 1 June and 15 August.




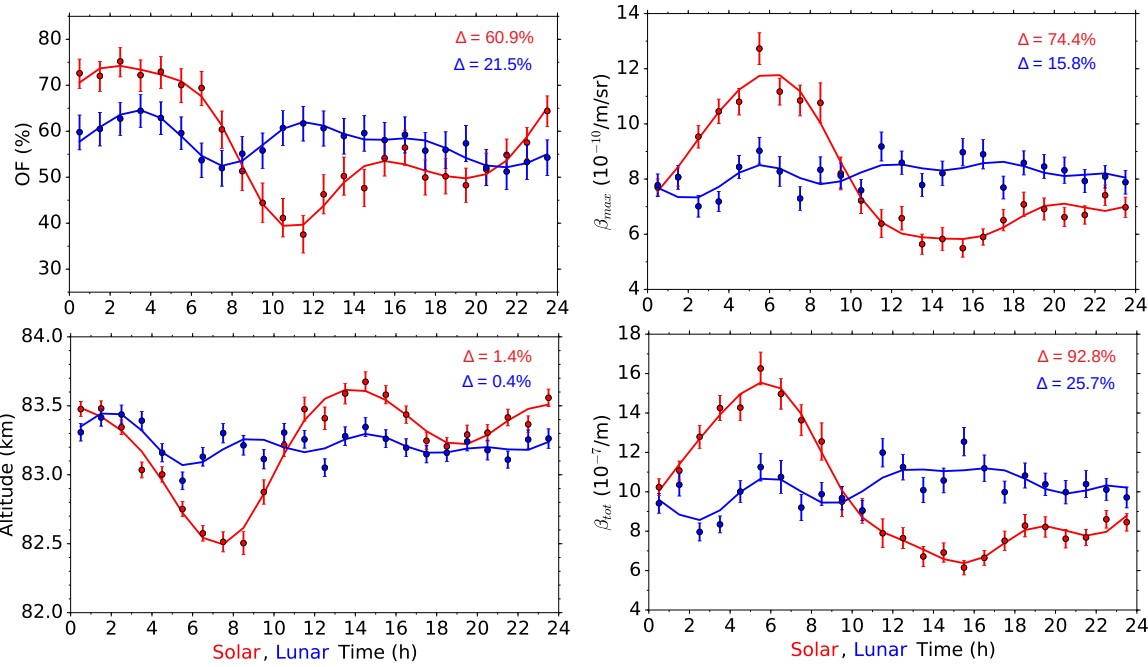

**Figure 2.** Mean solar (red) and lunar (blue) time variations of NLC occurrence frequency (OF), altitude and the maximum and integrated brightness ($\beta_{\max}$, $\beta_{\text{tot}}$) between 19 June and 29 July from 1997 to 2017. Symbols are hourly mean values and vertical bars errors of the means. Solid lines are harmonic fits with periods of 24, 12, 8 and 6 hours to the mean values. The relative variations of the fits over day $\Delta = (\max - \min)/\text{mean}$ are indicated.



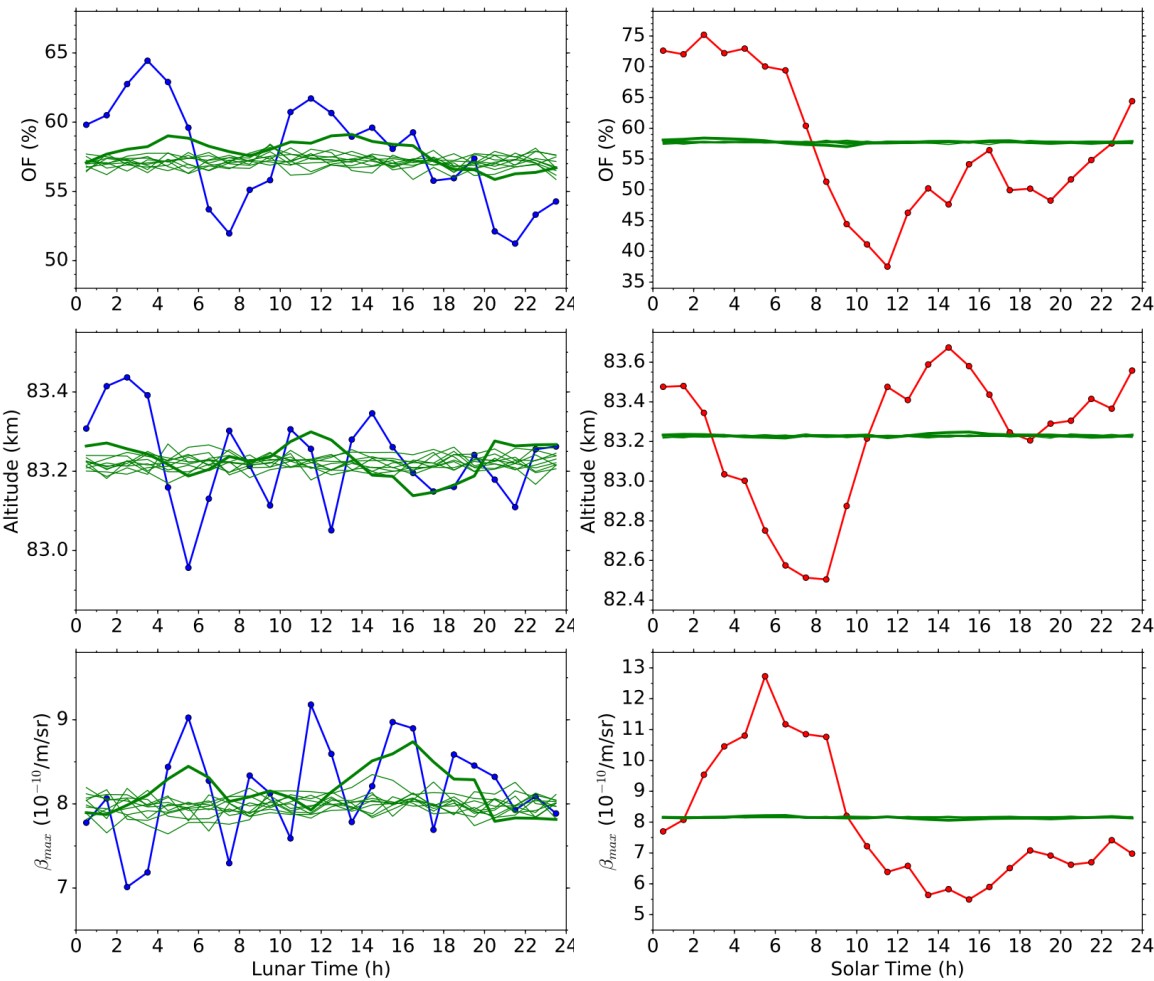

**Figure 3.** Simulated residual impact of solar onto lunar (left panels) and lunar onto solar (right panels) tidal variations, as introduced by the distribution of measurement times between 19 June and 29 July from 1997 to 2017. Method 1: actual times (thick green curves). Method 2: randomly generated times (thin green curves). Causative variations were reconstructed from harmonic fits with periods of 24, 12, 8 and 6 hours, see Table 1. For details see text. NLC parameters are: occurrence frequency (top), altitude (middle), brightness (bottom). Left panels: measured lunar dependence (blue), simulated impact from solar variations (green). Right panels: measured solar dependence (red), simulated impact from lunar variations (green).



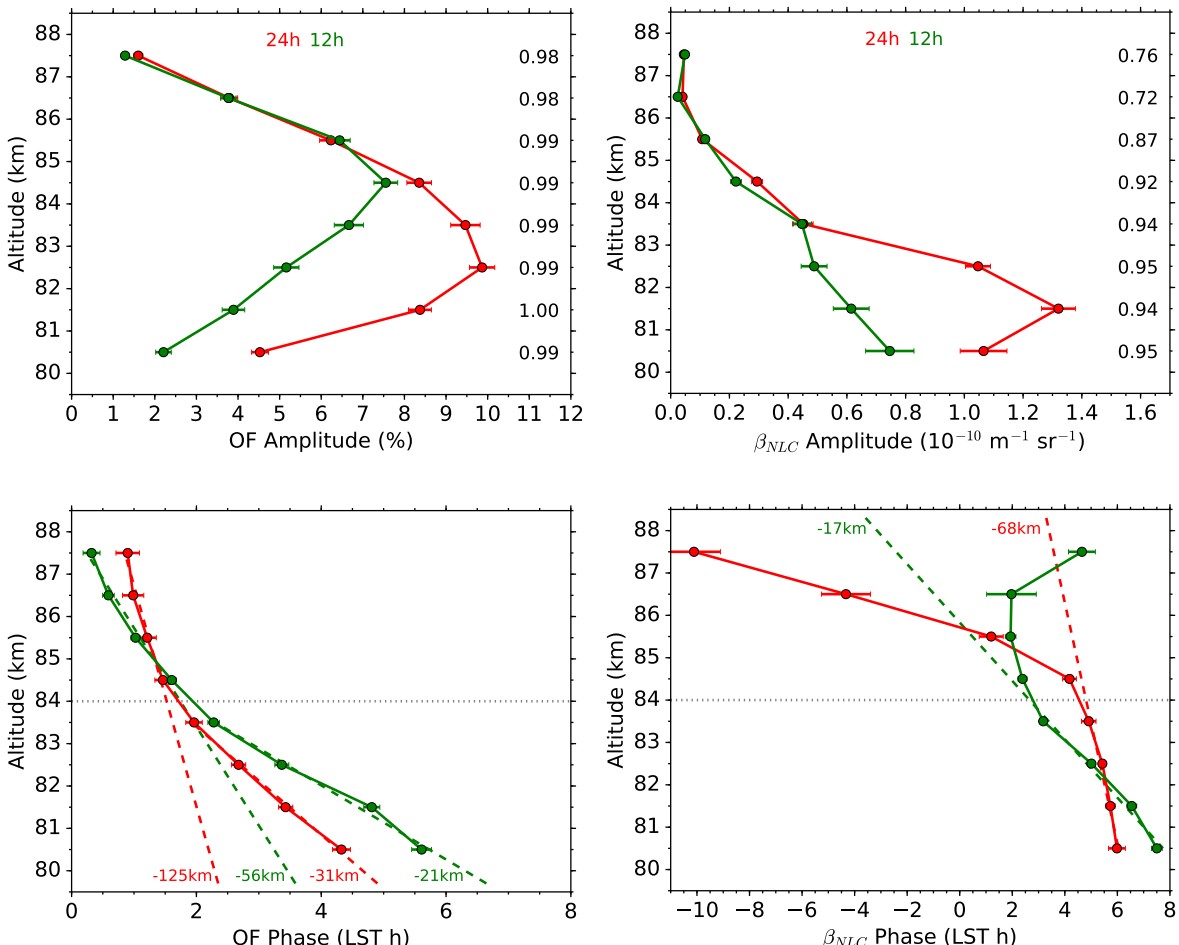

**Figure 4.** Mean solar amplitudes (top) and phases (bottom) of NLC occurrence frequency (OF, left) and brightness ($\beta_{\mathrm{NLC}}$, right) between 1 June and 15 August from 1997 to 2017. Diurnal components in red, semidiurnal components in green. Black numbers at the upper panels are correlation coefficients of the harmonic fits for the corresponding altitudes. Colored numbers at the lower panels are vertical wavelengths as calculated from the phase slopes (dashed lines), separated for altitudes below and above 84 km (gray dotted line).

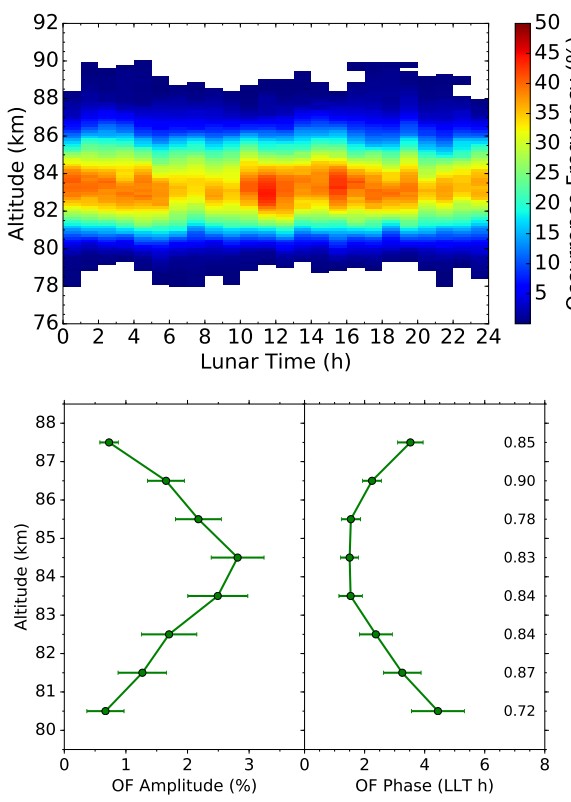

**Figure 5.** Top: Mean variations of NLC occurrence frequency with altitude and lunar time from 1997 to 2017. The plot contains 3450 hours of lidar measurements between 19 June and 29 July. Bottom: Semidiurnal amplitudes and phases determined from data of the top panel. Black numbers are correlation coefficients of the harmonic fits for the corresponding altitudes.



**Table 1.** Amplitudes (A) and phases (P) of solar and lunar tidal oscillations as determined from harmonic fits with periods of 24, 12, 8 and 6 hours to the data from 1997 to 2017. Amplitudes are given both in absolute and relative units. Absolute units (abs) are: occurrence frequency (OF) in %, altitude ($z_c$) in km, maximum brightness ($\beta_{max}$) in $10^{-10}\,m^{-1}\,sr^{-1}$, total brightness ($\beta_{tot}$) in $10^{-7}\,sr^{-1}$. Relative units (rel) are in % with respect to the mean value. Fit quality is given as correlation coefficient r.

| Parameter | Solar | | | | Lunar | | | |
|---|---|---|---|---|---|---|---|---|
| | OF | $z_c$ | $\beta_{max}$ | $\beta_{tot}$ | OF | $z_c$ | $\beta_{max}$ | $\beta_{tot}$ |
| $A_{24}$ (abs) | $13.40 \pm 1.23$ | $0.33 \pm 0.02$ | $2.54 \pm 0.14$ | $3.86 \pm 0.20$ | $1.68 \pm 1.16$ | $0.04 \pm 0.02$ | $0.34 \pm 0.14$ | $0.79 \pm 0.19$ |
| $A_{12}$ (abs) | $7.32 \pm 1.22$ | $0.33 \pm 0.02$ | $1.23 \pm 0.14$ | $1.56 \pm 0.20$ | $3.92 \pm 1.27$ | $0.08 \pm 0.02$ | $0.16 \pm 0.13$ | $0.21 \pm 0.17$ |
| $A_{08}$ (abs) | $2.92 \pm 1.25$ | $0.05 \pm 0.02$ | $0.22 \pm 0.14$ | $0.31 \pm 0.19$ | $2.68 \pm 1.23$ | $0.06 \pm 0.02$ | $0.19 \pm 0.14$ | $0.47 \pm 0.19$ |
| $A_{06}$ (abs) | $1.31 \pm 1.10$ | $0.03 \pm 0.02$ | $0.16 \pm 0.12$ | $0.37 \pm 0.19$ | $1.04 \pm 1.00$ | $0.07 \pm 0.02$ | $0.27 \pm 0.14$ | $0.47 \pm 0.19$ |
| $A_{24}$ (rel) | $23.46 \pm 2.16$ | $0.40 \pm 0.03$ | $31.86 \pm 1.76$ | $39.09 \pm 2.04$ | $2.91 \pm 2.01$ | $0.04 \pm 0.02$ | $4.21 \pm 1.78$ | $7.75 \pm 1.85$ |
| $A_{12}$ (rel) | $12.81 \pm 2.14$ | $0.40 \pm 0.03$ | $15.38 \pm 1.76$ | $15.80 \pm 1.93$ | $6.80 \pm 2.20$ | $0.10 \pm 0.03$ | $1.96 \pm 1.63$ | $2.09 \pm 1.72$ |
| $A_{08}$ (rel) | $5.11 \pm 2.18$ | $0.06 \pm 0.03$ | $2.74 \pm 1.75$ | $3.10 \pm 1.89$ | $4.64 \pm 2.14$ | $0.07 \pm 0.03$ | $2.37 \pm 1.69$ | $4.63 \pm 1.85$ |
| $A_{06}$ (rel) | $2.30 \pm 1.92$ | $0.04 \pm 0.03$ | $2.02 \pm 1.56$ | $3.76 \pm 1.93$ | $1.80 \pm 1.73$ | $0.08 \pm 0.03$ | $3.27 \pm 1.73$ | $4.59 \pm 1.85$ |
| $P_{24}$ | $1.83 \pm 0.36$ | $18.13 \pm 0.26$ | $4.60 \pm 0.21$ | $4.42 \pm 0.20$ | $7.37 \pm 3.58$ | $0.37 \pm 2.73$ | $14.16 \pm 1.74$ | $14.48 \pm 1.02$ |
| $P_{12}$ | $3.22 \pm 0.32$ | $0.66 \pm 0.13$ | $5.73 \pm 0.23$ | $5.23 \pm 0.24$ | $1.99 \pm 0.63$ | $0.94 \pm 0.51$ | $5.87 \pm 2.46$ | $4.41 \pm 2.66$ |
| $P_{08}$ | $6.70 \pm 0.55$ | $3.64 \pm 0.65$ | $5.11 \pm 0.93$ | $4.24 \pm 0.92$ | $2.56 \pm 0.62$ | $0.93 \pm 0.49$ | $5.05 \pm 1.23$ | $5.65 \pm 0.58$ |
| $P_{06}$ | $0.75 \pm 2.34$ | $3.92 \pm 0.75$ | $1.36 \pm 4.08$ | $0.37 \pm 2.00$ | $4.28 \pm 1.93$ | $1.91 \pm 0.31$ | $4.90 \pm 0.57$ | $5.13 \pm 0.44$ |
| r | 0.99 | 0.99 | 0.98 | 0.99 | 0.95 | 0.79 | 0.60 | 0.72 |