# Peer review of "Solar and lunar tides in noctilucent clouds as determined by ground-based lidar"

_Atmospheric Chemistry and Physics, 2018_

## Referee Comment (RC1) · Anonymous Referee #2 · 13 Jul 2018

SUMMARY

This paper extends other recent studies of lunar tide effects in noctilucent cloud (NLC) observations by using the long data record (21 years) of lidar measurements collected at the ALOMAR observatory in Norway. The lidar measurements supplement other satellite data sets because all local times are sampled, which improves the ability to separate solar and lunar tidal signatures. These data have been used successfully in many other studies of NLC behavior.

This paper is well-written, and the results are generally reasonable. Some suggestions and comments related to specific items are provided below.

[Figure]

**SPECIFIC COMMENTS**

1. p. 3, line 11: Please give 1-2 references for examples of the application of the superposed epoch analysis method.

2. p. 3, line 15: Extending the harmonic fit analysis to a 4th order term (period = 6 hours) requires high data quality to ensure that small noise fluctuations do not alias into apparent real behavior. Given the small magnitudes that are reported for this term in Table 1, is there any to demonstrate statistically that using it is valid?

3. p. 4, lines 16-21: The magnitude of the data reduction with the use of a "core" season is not that much different than the reduction when a long-term brightness limit is imposed (35% vs. 48%). It seems more likely that the brightness threshold eliminates some faint clouds that have a greater relative response to the weak lunar signal, whereas the use of a core season may actually improve the opportunity to identify this signal because faint clouds have better background conditions in which to form. If the authors agree with this premise, I suggest adding it to the discussion.

4. p. 5, lines 7-8: Fiedler et al. [2011] show significant interannual variation in amplitude and phase of solar tidal components in NLC properties measured at ALOMAR over 14 years. If similar variations are present in lunar tidal behavior, does that cause a problem for the application of the superposed epoch analysis method, which combines data taken during many separate years?

5. p. 5, lines 21-22: Note that von Savigny et al. [2017] and Hoffman et al. [2018] only present a semi-diurnal variation (as a single fit). So you should be careful in evaluating the agreement (or difference) between amplitudes and phases derived from those analyses vs. the 4-term results presented here. I would not expect complete agreement even if the same data set was examined because of the extra terms present in the 4-term fit.

6. p. 6, lines 12-14: Some of the terms in the lunar tidal results are barely larger

than their 1-sigma uncertainty (e.g. 6-hour period for OF, 12-hour and 8-hour period for Bmax). Since the original lunar signal is fairly weak, are you sure that all of these terms are really significant? Have you looked at fit results using only 2 or 3 terms?

7. p. 7, lines 28-30: What is the meaning of a negative vertical wavelength? Is this related to the sign of the phase term?

8. p. 8, lines 30-32: Would you expect that a nadir-viewing instrument (such as SBUV), that integrates the NLC signal vertically, would see something like a linear sum of the frequency values at each altitude? Or would it see a weighted sum because the larger (brighter) ice particles are present at lower altitudes?

TYPOGRAPHICAL ERRORS AND GRAMMATICAL SUGGESTIONS

p. 1, line 21: "waver" should be "water".

p. 2, line 2: "Prominent influence have diurnal and semidiurnal components" could be changed to "Diurnal and semidiurnal components have a prominent influence".

p. 2, line 6: "in atmosphere" should be "in the atmosphere".

p. 2, line 18: "overhead" could be changed to "above".

p. 2, line 31: "begin of" should be "the beginning of".

p. 3, line 1: "a subset of 3100 hours" could be changed to "with a subset of 3100 hours that".

p. 4, line 19: "restricts to" could be changed to "restricts the sampling to".

p. 4, line 28: "observed 1997" should be "observed in 1997".

p. 5, line 27: "Limp" should be "Limb".

p. 6, line 8: "make aware" could be changed to "make the reader aware".

p. 6, line 28: "results into" could be changed to "results in".

p. 6, line 34: "only be small impacted" could be changed to "have only a small impact".

p. 9, line 21: "vertical" should be "vertically".

p. 9, line 22: "hinting for" could be changed to "suggesting".
* * *

---

## Referee Comment (RC2) · Anonymous Referee #1 · 16 Jul 2018

GENERAL COMMENTS

The present paper analyses a long-term time series of ground-based lidar data to derive properties of solar and lunar tides in noctilucent clouds. The properties of these weak features are difficult to extract out of all other components of atmospheric variability. Hence, long-term datasets are needed to be able to apply statistical methods. The present paper introduces a new and comparatively long dataset to this kind of research. This ground-based dataset has properties, which are complementary to those of the satellite datasets used before. It provides data for only one location but with a much better temporal resolution, which allows for a direct identification of tides on timescales of

hours to one day. Different properties of solar and lunar tides are successfully derived and, when possible, compared to previous results from other measurement techniques, showing agreement in many aspects. There are not many independent derivations of these properties in the literature and the comparison with this complementary dataset is very valuable. The paper is well-written and easy to understand. It fits well to the scope of ACP so that I recommend the publication after addressing some minor points listed below.

SPECIFIC COMMENTS

Page 2, line 22: "however, it takes one month to cover all lunar times": It could be helpful for the reader to explain this by 2 or 3 more sentences.

Page 3, line 2: "the mean probability to observe NLC at this location is âĹij48 %": Is this information really useful? The reference of 6400 measurement hours appears to be quite arbitrary, as it depends on weather conditions, etc. One could use the total number of possible measurement hours within the 21 seasons, but still I am not sure if this information is needed. Please explain or change accordingly.

Page 3, lines 5-8: "For details the reader is referred to Fiedler et al. (2009)": I would find it useful to add some more sentences on the calculation method, although Fiedler et al. (2009) is referenced. This would make the article more self-contained and help to directly understand it for readers without a strong background in lidar remote sensing.

Page 3, line 15: "to the hourly mean values": It might not directly be clear that the mean values from the epoch averaging are meant in contrast to, e.g., an hourly averaged time series (other approaches fit the sinusoidal functions directly to time series). Using other words like "epoch averages" could make this clearer.

Page 3, line 17: "The mean NLC parameters are randomly diversified within their error bars (1000 times for each hour)": Which distribution function is used for this (e.g., are the random samples normally distributed with the standard deviation set to the error

bars or are they uniformly distributed with sharp edges at the error bar range)?

Page 3, line 23 and Figure 1, caption: "The plots contain 6400 hours of lidar measurements": Is this really the average over all available measurement hours or the average over the 3100 hours, which contain NLC? If 6400 hours is correct: Is it useful to integrate the measurements without NLC in this plot? To my understanding, these measurements are not used in the rest of the analysis and I would have expected to apply the same filters as for the analysis here.

Page 4, line 22: "The plots contain 3450 measurement and 2030 NLC hours from 1997 to 2017": a similar question here: Are the measurement hours without NLC averaged into to mean data? Or what does the differencing between measurement and NLC hours actually mean?

Page 5, line 1: "most intense for occurrence frequency": According to the Delta-value in the plots, the total backscatter coefficient varies most. So, to which value is this statement related?

Page 5, line 19ff: You compare your values of the semidiurnal amplitudes A12 to those of the satellite studies. As you mention also in the introduction, it takes about one month for sun-synchronous satellites to sample all local lunar times. Hence, the satellites strictly observe a superposition of a semidiurnal and a semimonthly lunar tide and it is necessary for the interpretation to assume that the semidiurnal tide dominates over the semimonthly tide. This assumption is commonly made, but is sometimes still under debate. Could you comment on, first, to what extent the good matching of your results with the satellite results support this assumption and, second, if it could be possible in future to also extract the semimonthly tide from your data in order to cleanly separate both?

Page 6, Line 13: "the observed lunar tidal behavior, different for the NLC parameters": the phrasing is not very clear to me. Maybe the sentence could be restructured to put emphasis on the "different" and not on "all...contribute".

Page 8, Line 33: "The altitude structure of the lunar semidiurnal tide in layered phenomena of the summer mesopause region was never studied so far": To my understanding, this is not true in this generality. E.g., von Savigny et al., 2017 shows the altitude structure of the semidiurnal lunar tide in MLS temperature up to 90km altitude and also Hoffmann et al., 2018 shows the altitude dependence of the semidiurnal lunar tide in several NLC related parameters. However, I agree that the phase progression has not been quantified and discussed in these studies as the authors do it here.

TECHNICAL CORRECTIONS

Page 1, line 25: "even when epoch averaging over many years and where attributed to impacts of atmospheric thermal tides": A bit hard to understand. First, maybe: "when epoch averaging over many years is applied" and second, "where" has probably to be "were". Page 4, line 22: "The plots contain 3450 measurement", add "s" to measurement

REFERENCES

Hoffmann, C. G., von Savigny, C., Hervig, M. E., & Oberbremer, E. (2018). The lunar semidiurnal tide at the polar summer mesopause observed by SOFIE. Journal of Atmospheric and Solar-Terrestrial Physics, 167, 134–145. https://doi.org/10.1016/j.jastp.2017.11.014

von Savigny, C., DeLand, M. T., & Schwartz, M. J. (2017). First identification of lunar tides in satellite observations of noctilucent clouds. Journal of Atmospheric and Solar-Terrestrial Physics, 162(Supplement C), 116–121. https://doi.org/10.1016/j.jastp.2016.07.002

---

## Author Comment (AC1) · 17 Oct 2018

We appreciate the comments from the reviewer and have taken the suggestions into account. In the following we respond to the remarks point by point. Our responses are in italics. Line numbers refer to the revision with changes marked. Changes in the manuscript have been marked in the following way: deleted text in red, new text in blue.
* * *
**Anonymous Referee #2**
()
SUMMARY
This paper extends other recent studies of lunar tide effects in noctilucent cloud (NLC) observations by using the long data record (21 years) of lidar measurements collected at the ALOMAR observatory in Norway. The lidar measurements supplement other satellite data sets because all local times are sampled, which improves the ability to separate solar and lunar tidal signatures. These data have been used successfully in many other studies of NLC behavior.
This paper is well-written, and the results are generally reasonable. Some suggestions and comments related to specific items are provided below.

SPECIFIC COMMENTS
1. p. 3, line 11: Please give 1-2 references for examples of the application of the superposed epoch analysis method.

*We have referenced the first application of this method by Chree, 1912: page 3, line 20.*

2. p. 3, line 15: Extending the harmonic fit analysis to a 4th order term (period = 6 hours) requires high data quality to ensure that small noise fluctuations do not alias into apparent real behavior. Given the small magnitudes that are reported for this term in Table 1, is there any to demonstrate statistically that using it is valid?

*6 hour periods are frequently found in lidar observation, e.g. Fricke-Begemann and Höffner [2005], which is the main reason for including the $4^{th}$ order term in our work. We find the data quality of the solar time series sufficient for this (correlation coefficients between fit and data are close to 1, see Table 1), and use this term for the lunar time series for homogeneity. As you see we have not scientifically interpreted the 6-hour results. On the other hand it appears useful to show the result for 2 reasons. At first it demonstrates that these short periods are (mostly) of small magnitude and do barely contribute to the overall tidal variations. At second it is important to include them for the simulations of the mutual residual impact between solar and lunar tidal oscillations (reliability of tidal parameters). See also point 6 below.*

3. p. 4, lines 16-21: The magnitude of the data reduction with the use of a "core" season is not that much different than the reduction when a long-term brightness limit is imposed (35% vs. 48%). It seems more likely that the brightness threshold eliminates some faint clouds that have a greater relative response to the weak lunar signal, whereas the use of a core season may actually improve the opportunity to identify this signal because faint clouds have better background conditions in which to form. If the

authors agree with this premise, I suggest adding it to the discussion.

*The long-term brightness limit (BETAmax > 4) removes all faint clouds (1 < BETAmax < 4) which results into completely separated data sets. We also investigated these 2 data sets and found lunar tidal signatures in both of them. Also, the seasonal dependence of faint clouds occurrence is smaller compared to that of brighter clouds, cf. Schmidt et al. [2018] (Fig. 3), which is caused by the seasonal variation of temperature and water vapor. This suggests that faint clouds are not primarily responsible for lunar tidal signatures in the overall NLC population.*
*We did a small text adjustment: page 4, lines 27-28.*

4. p. 5, lines 7-8: Fiedler et al. [2011] show significant interannual variation in amplitude and phase of solar tidal components in NLC properties measured at ALOMAR over 14 years. If similar variations are present in lunar tidal behavior, does that cause a problem for the application of the superposed epoch analysis method, which combines data taken during many separate years?

*Such variations would certainly impact the results extracted from this method. We have tried to investigate this topic by splitting the data set (first and last half of the time series). However, despite our large database covering 21 years it seems to be still too small for such investigations.*

5. p. 5, lines 21-22: Note that von Savigny et al. [2017] and Hoffman et al. [2018] only present a semi-diurnal variation (as a single fit). So you should be careful in evaluating the agreement (or difference) between amplitudes and phases derived from those analyses vs. the 4-term results presented here. I would not expect complete agreement even if the same data set was examined because of the extra terms present in the 4-term fit.

*In principle we agree, but comparing fits and original data at Fig. 2 in von Savigny et al. [2017] suggest barely other periods than a semi-diurnal variation. So, comparing their 12-hour components with ours should be appropriate. See also next point.*

6. p. 6, lines 12-14: Some of the terms in the lunar tidal results are barely larger than their 1-sigma uncertainty (e.g. 6-hour period for OF, 12-hour and 8-hour period for Bmax). Since the original lunar signal is fairly weak, are you sure that all of these terms are really significant? Have you looked at fit results using only 2 or 3 terms?

*Yes, indeed we calculated all fit results not only for the 4-term version but also for only 2 terms (24- and 12-hour periods) in order to check the influence of the additional terms. It turned out that the fit algorithm is fairly robust, the amplitudes and phases resulting from both fit versions are close to each other. For example, for the 12-hour component of Bmax the deviations between including/omitting 8-hour and 6-hour periods are: 1.7% for amplitude value, 1.0% for amplitude error, 8.7% for phase value, 3.9% for phase error.*
*We added this information to the text: page 5, lines 18-22.*

7. p. 7, lines 28-30: What is the meaning of a negative vertical wavelength? Is this related to the sign of the phase term?

*Yes, the phase is decreasing with increasing altitude.*

8. p. 8, lines 30-32: Would you expect that a nadir-viewing instrument (such as SBUV), that integrates the NLC signal vertically, would see something like a linear sum of the frequency values at each altitude? Or would it see a weighted sum because the larger (brighter) ice particles are present at lower altitudes?

*Good question …, we would expect that such instrument would see a weighted sum because the vertical integration of the scattered light should be dominated by the brighter ice particles. Compared to the lidar, the SBUV signal has a weaker dependence on particle size (r^3 versus r^6) which should result in a weaker weighting. Additionally the anti-correlation between number density and size of the particles might further weakening the weighting.*

TYPOGRAPHICAL ERRORS AND GRAMMATICAL SUGGESTIONS
p. 1, line 21: "waver" should be "water".
p. 2, line 2: "Prominent influence have diurnal and semidiurnal components" could be changed to "Diurnal and semidiurnal components have a prominent influence".
p. 2, line 6: "in atmosphere" should be "in the atmosphere".
p. 2, line 18: "overhead" could be changed to "above".
p. 2, line 31: "begin of" should be "the beginning of".

*All done.*

p. 3, line 1: "a subset of 3100 hours" could be changed to "with a subset of 3100 hours that".

*Already changed due to comments by the other reviewer.*

p. 4, line 19: "restricts to" could be changed to "restricts the sampling to".
p. 4, line 28: "observed 1997" should be "observed in 1997".
p. 5, line 27: "Limp" should be "Limb".
p. 6, line 8: "make aware" could be changed to "make the reader aware".
p. 6, line 28: "results into" could be changed to "results in".

*All done.*

p. 6, line 34: "only be small impacted" could be changed to "have only a small impact".

*We leave it like it is.*

p. 9, line 21: "vertical" should be "vertically".

*Already changed due to comments by the other reviewer.*

p. 9, line 22: "hinting for" could be changed to "suggesting".

*Done.*

*References:*

*Fricke-Begemann, C., and J. Höffner (2005), Temperature tides and waves near the mesopause from lidar observations at two latitudes, J. Geophys. Res., 110, D19103, doi:10.1029/2005JD005770.*

*Schmidt, F., Baumgarten, G., Berger, U., Fiedler, J., Lübken, F.-J. (2018), Local time dependence of polar mesospheric clouds: a model study, Atmos. Chem. Phys.,18, 8893–8908, doi:10.5194/acp-18-8893-2018.*

---

## Author Comment (AC2) · 17 Oct 2018

We appreciate the comments from the reviewer and have taken the suggestions into account. In the following we respond to the remarks point by point. Our responses are in italics. Line numbers refer to the revision with changes marked. Changes in the manuscript have been marked in the following way: deleted text in red, new text in blue.
* * *
**Anonymous Referee #1**
()
GENERAL COMMENTS
The present paper analyses a long-term time series of ground-based lidar data to derive properties of solar and lunar tides in noctilucent clouds. The properties of these weak features are difficult to extract out of all other components of atmospheric variability. Hence, long-term datasets are needed to be able to apply statistical methods. The present paper introduces a new and comparatively long dataset to this kind of research. This ground-based dataset has properties, which are complementary to those of the satellite datasets used before. It provides data for only one location but with a much better temporal resolution, which allows for a direct identification of tides on timescales of hours to one day. Different properties of solar and lunar tides are successfully derived and, when possible, compared to previous results from other measurement techniques, showing agreement in many aspects. There are not many independent derivations of these properties in the literature and the comparison with this complementary dataset is very valuable. The paper is well-written and easy to understand. It fits well to the scope of ACP so that I recommend the publication after addressing some minor points listed below.

SPECIFIC COMMENTS
Page 2, line 22: "however, it takes one month to cover all lunar times": It could be helpful for the reader to explain this by 2 or 3 more sentences.

*We have added text to clarify this relationship: page 2, lines 21-27.*

Page 3, line 2: "the mean probability to observe NLC at this location is âĹij48 %": Is this information really useful? The reference of 6400 measurement hours appears to be quite arbitrary, as it depends on weather conditions, etc. One could use the total number of possible measurement hours within the 21 seasons, but still I am not sure if this information is needed. Please explain or change accordingly.

*We find it indeed useful to bring up these numbers. First of all they indicate the mean state (here: NLC occurrence frequency) before variability is discussed later in the manuscript. They additionally show that the tracer for our investigation of tides should allow statistically robust conclusions because NLC are no seldom events at this location. Last but not least this large number of measurement hours is quite unique for a ground-based lidar at this geographic location (full sunlight during summer etc.). For these reasons we leave the numbers but adjusted the text for improved readability: page 3, lines 4-5.*

Page 3, lines 5-8: "For details the reader is referred to Fiedler et al. (2009)": I would find it useful to add some more sentences on the calculation method, although Fiedler

et al. (2009) is referenced. This would make the article more self-contained and help to directly understand it for readers without a strong background in lidar remote sensing.

*We have added text according to the suggestion: page 3, lines 8-17.*

Page 3, line 15: "to the hourly mean values": It might not directly be clear that the mean values from the epoch averaging are meant in contrast to, e.g., an hourly averaged time series (other approaches fit the sinusoidal functions directly to time series). Using other words like "epoch averages" could make this clearer.

*Done: page 3, line 25.*

Page 3, line 17: "The mean NLC parameters are randomly diversified within their error bars (1000 times for each hour)": Which distribution function is used for this (e.g., are the random samples normally distributed with the standard deviation set to the error bars or are they uniformly distributed with sharp edges at the error bar range)?

*The random samples are uniformly distributed within the error bar range, which results in a larger error bar compared to the Gaussian weighted random distribution.*

Page 3, line 23 and Figure 1, caption: "The plots contain 6400 hours of lidar measurements": Is this really the average over all available measurement hours or the average over the 3100 hours, which contain NLC? If 6400 hours is correct: Is it useful to integrate the measurements without NLC in this plot? To my understanding, these measurements are not used in the rest of the analysis and I would have expected to apply the same filters as for the analysis here.

*The phrasing is something inexplicit, the 6400 hours apply for the occurrence frequency, for brightness only times with NLC detections are valid (3100 hours).*
*We corrected the text accordingly: page 3, line 33 – page4, line 1 ; Figure 1 caption.*

Page 4, line 22: "The plots contain 3450 measurement and 2030 NLC hours from 1997 to 2017": a similar question here: Are the measurement hours without NLC averaged into to mean data? Or what does the differencing between measurement and NLC hours actually mean?

*We think that rephrasing the text (see above) allows the reader to understand the numbers now.*

Page 5, line 1: "most intense for occurrence frequency": According to the Delta-value in the plots, the total backscatter coefficient varies most. So, to which value is this statement related?

*You are right, we meant the match between measurement and fit. Text is removed: page 5, lines 11-13.*

Page 5, line 19ff: You compare your values of the semidiurnal amplitudes A12 to those of the satellite studies. As you mention also in the introduction, it takes about one month for sun-synchronous satellites to sample all local lunar times. Hence, the satellites strictly observe a superposition of a semidiurnal and a semimonthly lunar tide and it is necessary for the interpretation to assume that the semidiurnal tide dominates over the semimonthly tide. This assumption is commonly made, but is sometimes still under debate. Could you comment on, first, to what extent the good matching of your results with the satellite results support this assumption and, second, if it could be possible in future to also extract the semimonthly tide from your data in order to cleanly separate both?

*From the match of both results one could deduce a dominating semidiurnal component. Basically our data allow such investigations, which will be subject of future work. Nevertheless we added text to make the reader aware of this topic: page 6, lines 4-6.*

Page 6, Line 13: "the observed lunar tidal behavior, different for the NLC parameters": the phrasing is not very clear to me. Maybe the sentence could be restructured to put emphasis on the "different" and not on "all. . .contribute".

*Done: page 6, lines 29-30.*

Page 8, Line 33: "The altitude structure of the lunar semidiurnal tide in layered phenomena of the summer mesopause region was never studied so far": To my understanding, this is not true in this generality. E.g., von Savigny et al., 2017 shows the altitude structure of the semidiurnal lunar tide in MLS temperature up to 90km altitude and also Hoffmann et al., 2018 shows the altitude dependence of the semidiurnal lunar tide in several NLC related parameters. However, I agree that the phase progression has not been quantified and discussed in these studies as the authors do it here.

*We changed the text to be more specific: page 9, lines 13-14; page 10, line 4.*

TECHNICAL CORRECTIONS
Page 1, line 25: "even when epoch averaging over many years and where attributed to impacts of atmospheric thermal tides": A bit hard to understand. First, maybe: "when epoch averaging over many years is applied" and second, "where" has probably to be "were".

*Done.*

Page 4, line 22: "The plots contain 3450 measurement", add "s" to measurement

*Both numbers are in units of hours: "3450 measurement and 2030 NLC hours". We changed the text for a better understanding: page 4, lines 32-33.*